# Performance Comparison of Genomic Best Linear Unbiased Prediction and Four Machine Learning Models for Estimating Genomic Breeding Values in Working Dogs

**DOI:** 10.3390/ani15030408

**Published:** 2025-02-02

**Authors:** Joseph A. Thorsrud, Katy M. Evans, Kyle C. Quigley, Krishnamoorthy Srikanth, Heather J. Huson

**Affiliations:** 1Department of Animal Sciences, College of Agriculture and Life Sciences, Cornell University, 201 Morrison Hall, 507 Tower Road, Ithaca, NY 14853, USA; jat325@cornell.edu (J.A.T.); hjh3@cornell.edu (H.J.H.); 2The Seeing Eye Inc., 1 Seeing Eye Wy, Morristown, NJ 07960, USA; kevans@seeingeye.org (K.M.E.);; 3School of Veterinary Medicine and Science, University of Nottingham, Sutton Bonington, Loughborough LE12 5RD, UK

**Keywords:** genomic prediction, breeding values, machine learning models, dog breeding, genetic selection, genomic best linear unbiased prediction, random forest, support vector machine, extreme gradient boosting, multilayer perceptron

## Abstract

This study aims to improve the breeding of guide dogs by using genetic information to predict important health and behavior traits. Guide dogs need to be healthy and attentive to effectively assist people with visual impairments. This study compares several methods for predicting whether a dog might develop certain health issues, such as anodontia (missing teeth), distichiasis (extra eyelashes that can irritate the eyes), or oral papillomatosis (oral tumors caused by a virus), as well as behaviors like high distractibility, based on their genetic makeup. Data from German Shepherds, Golden Retrievers, Labrador Retrievers, and their crosses were analyzed to see which prediction methods work best given different models and data parameters. The results show that all the tested methods were similarly effective in predicting these traits. Notably, simpler and less time-intensive methods and data collection processes perform just as well as more complex ones. This means that dog breeders can use these genetic prediction tools without investing in expensive technology or genetic testing. By applying these methods, breeders can make better informed decisions when selecting dogs for breeding, focusing on those more likely to be healthy and exhibit desirable behaviors. Ultimately, this approach can lead to the development of healthier and more capable guide dogs, benefiting individuals who rely on them and contributing to the overall well-being of the dog population.

## 1. Introduction

Modern dog breeding took shape in the mid-19th century with a focus on enhancing phenotypic attributes and aligning them with esthetic ideals and functional roles established by breed standards [1]. The use of phenotypes and pedigree data to track lineages allowed breeders to direct selection toward their desired attributes. Modern updates to pedigree-based selection allow for the quantification of estimated breeding values (EBVs). EBVs predict the genetic potential of an individual to pass on traits to offspring by combining pedigree and phenotypic data. Unlike tests that simply identify if an animal is a carrier of a specific gene, EBVs provide a comprehensive assessment of an animal’s genetic merit for quantitative traits, enabling breeders to make more informed selection decisions and improve traits across generations. Notable successes of this technique include the estimation of hip and elbow dysplasia breeding values in dogs, leading to improved joint health [2,3,4,5]. However, compared to agricultural species, the implementation of EBVs in canine breeding programs has lagged due to challenges such as smaller reference populations, less standardized selection criteria, and limited collaboration among breeders. This gap highlights the need for research focusing on optimizing genetic prediction methods for dogs, which is the aim of the present study.

As these methods are refined, it is crucial to ensure that breeding decisions prioritize the welfare of individual animals, preserve genetic diversity, and mitigate the perpetuation of harmful traits. Overemphasis on narrowly defined breed-specific characteristics can exacerbate existing hereditary problems, raising ethical concerns around breeding practices that may compromise long-term health and well-being [6]. Researchers and breeders must remain vigilant in balancing performance or esthetic goals with robust welfare standards, including the responsible use of EBVs. By employing breeding values to identify and minimize hereditary disorders and maintaining balanced trait selection, breeders can harness these tools ethically and effectively to select for or against traits [7]. Though systemic problems arise with dog breeding, through a concerted focus on health and behavior traits, responsible breeding can avoid pitfalls that have plagued the field [8]. In this way, the primary focus remains on safeguarding and enhancing the quality of life of dogs while improving genomic tool development.

The introduction of genomic data-driven selection marked a transformative shift in breeding practices for many agricultural species [9]. Genomic selection utilizes DNA marker data to provide a more precise assessment of an individual’s genetic merit compared to traditional pedigree-based methods. This advancement took off with the usage of single-nucleotide polymorphisms (SNPs) for genotyping. By analyzing SNP markers and using them to infer regions in linkage, breeders gain deeper insights into the genetic composition of individuals, thus improving the accuracy and efficiency of breeding decisions [10]. The success of genomic selection has been particularly pronounced in agricultural species like dairy cattle, where it has led to significant advancements in milk production, health, and fertility [9,11]. One of the key benefits of genomic selection is the ability to identify an individual’s genetic merit at birth and differentiate between littermates, reducing the generation interval and accelerating the rate of genetic gain through selection compared to pedigree-based methods. These agricultural examples highlight the importance of large reference populations and consistently agreed-upon phenotypes among breeders for selection.

Several advanced genomic prediction models, including machine learning techniques, have been developed to leverage genomic data effectively. Machine learning models are particularly appealing due to their ability to model complex, nonlinear relationships in large datasets, which is advantageous in genomic prediction. Among traditional methods, Genomic Best Linear Unbiased Prediction (GBLUP) extends the traditional Best Linear Unbiased Prediction (BLUP) model by replacing the pedigree-based relationship matrix with a genomic relationship matrix derived from SNP markers [12]. This approach captures genetic relationships with greater accuracy, especially in scenarios where pedigree data are incomplete or unavailable. GBLUP is valued for its robustness and its ability to handle extensive datasets with numerous genetic markers, making it a widely adopted tool in modern breeding programs.

The first of the five machine learning models tested is Random Forest (RF), which represents a powerful model that has been previously used in genomic prediction [13,14,15]. This ensemble learning method constructs multiple decision trees during training and aggregates their predictions to enhance accuracy and mitigate overfitting. RF excels in managing complex interactions among multiple genes and demonstrates robustness against overfitting. The strengths of RF may make it particularly beneficial for predicting traits influenced by complex genetic factors. Support Vector Machine (SVM) is a machine learning technique designed for classification tasks that has been used in genomic selection before [14,16,17]. It identifies the hyperplane that best separates different classes in a high-dimensional space, making it well suited for genomic prediction where data dimensionality is high and traits are binary. Extreme Gradient Boosting (XGB) is an advanced implementation of gradient boosting algorithms. XGB builds models sequentially, with each new model aiming to correct the errors of its predecessors. This approach is highly efficient and effective at managing large, complex datasets, capturing intricate genetic patterns and potentially improving the accuracy of breeding value predictions. XGB’s capability to handle complex trait architectures makes it an already utilized tool in genomic prediction [17,18]. Multilayer Perceptron (MLP) comprises neural network models inspired by the structure and function of the human brain. MLPs consist of multiple layers of interconnected nodes that can model complex, nonlinear relationships within the data. Although MLPs require substantial computational resources and large datasets for training, they have shown promise in capturing complex genetic interactions and providing accurate predictions for traits with intricate genetic architectures [19,20].

The effectiveness of these genomic prediction models is closely tied to the quality and quantity of the genomic data available. SNP markers are critical data points used in these models, offering detailed insights into genetic variation across the genome. The density of SNP markers—referring to the number of markers analyzed—can potentially impact the accuracy of predictive models [21,22,23]. Generally, higher marker density improves prediction accuracy by capturing a greater portion of the genetic variation associated with traits. However, an excessive number of markers can lead to overfitting, where the model becomes too specialized to the training data and performs poorly on new data during validation, can increase computation time, or is cost-prohibitive [24]. Therefore, finding an optimal marker density is crucial for balancing predictive accuracy with practical considerations such as cost and computational capacity.

The focus of this study is on specific traits that vary in their epidemiology, heritability, and case count: anodontia, distichiasis, oral papillomatosis, and distraction. These traits were selected to provide insight into how heritability estimates and the number of cases can influence model performance. They also represent health and behavioral concerns with tangible ramifications for both dogs and breeders. Preventing painful ocular irritations (distichiasis) and health complications (anodontia and papillomatosis) helps maintain dogs’ well-being. Ensuring reliable working performance (distraction) can also minimize the number of dogs that need to be rehomed due to behavioral failures. Anodontia, a congenital absence of teeth, can lead to severe oral health issues and difficulties with feeding and has variability in presentation, cause, and severity [25]. Distichiasis, characterized by the presence of extra eyelashes that may grow towards the eye, can cause corneal irritation and impaired vision [26]. In other breeds, heritability has been estimated to be high, ranging from 0.276 to 0.720, indicating a potentially strong genetic component to the disease [27,28]. Oral papillomatosis, caused by the canine papillomavirus, results in benign oral tumors that, while generally non-life-threatening, can cause significant discomfort and interfere with eating [29,30]. As it is a viral infection, genetic factors may influence susceptibility, but environmental exposure plays a significant role, suggesting a lower heritability [29]. Distraction, a behavioral trait affecting a dog’s focus, is crucial for guide dogs whose effectiveness depends on their attentiveness [31]. Behavioral traits often have moderate heritability and can be more subjective in assessment but selection for focus and non-distracted behavior has been identified as an important aspect of guide dog breeding [32].

This study aims to compare the performance of genomic prediction models—GBLUP, RF, SVM, XGB, and MLP—in predicting breeding values for three binary health traits and one behavioral trait in a population of guide dogs. By evaluating these models across different breeds—including Labrador Retrievers, Golden Retrievers, German Shepherds, and Labrador and Golden Retriever crosses—the research aims to illuminate model performance under various population strategies. This includes both within-breed and across-breed analyses and takes into account different reference population sizes. Additionally, this study assesses model performance across traits with different heritabilities and a behavioral trait, as well as the impact of SNP marker density on predictive accuracy, offering a deeper understanding of how different models handle breed-specific genetic variations and trait characteristics.

The implications of this study extend beyond guide dogs, potentially influencing breeding programs for other populations of animals with smaller reference populations and unique characteristics that differentiate them from research conducted primarily on agricultural species. Improved predictive accuracy can lead to healthier, more capable working dogs and contribute to more effective and informed breeding decisions. By addressing the challenges unique to canine breeding, such as smaller population sizes and less standardized phenotyping, this study contributes valuable knowledge toward closing the gap in the application of genomic selection in dogs compared to agricultural species.

## 2. Materials and Methods

### 2.1. Population

This study utilized a dataset provided by The Seeing Eye organization encompassing the 26-year period of 1998–2024. The Seeing Eye breeds and trains guide dogs, including German Shepherds, Golden Retrievers, Labrador Retrievers, and Labrador and Golden Retriever crosses. The crosses are subsequently bred back to Labrador Retrievers. The crosses between Labradors and Golden Retrievers were originally introduced to combine the well-known temperament and working abilities of these breeds. Subsequent backcrossing to Labradors was employed to maintain a consistent appearance aligned with The Seeing Eye’s working requirements while also aiming to enrich genetic diversity. Phenotypic data were collected during the puppy and young dog raising and training phases before their placement with a visually impaired individual or selection for breeding. Phenotypic data were collected by The Seeing Eye from birth until approximately 4.5 years of age. The dataset encompasses both successful guide dogs and those identified as failures. Disease traits were recorded by an in-house veterinary team, while behavioral traits, specifically distractibility, were assessed by trainers. Models were run on the all-breed data, then within the breeds of Labrador Retrievers (LRs), Golden Retrievers (GRs), and German Shepherds (GSs) and combining the more similar breeds and crosses of Labrador Retrievers, Golden Retrievers, and their crosses (LR/GR). Animals with incomplete phenotypic or genotypic data were excluded from the analysis. The sample size included 2020 dogs for the health traits and 1290 dogs for the behavioral trait.

### 2.2. The Seeing Eye Phenotypic Dataset

Health phenotypes were diagnosed through physical examinations conducted at specific developmental stages. These included a puppy physical around 5 weeks of age (or slightly older if the puppy was purchased), a pre-training physical (PTP) at approximately 14–16 months, a pre-breeder physical about 3 months after the PTP, and a pre-class physical roughly 4 months after the PTP. The Seeing Eye uses a coding system with defined diagnostic criteria for each trait to standardize health trait characterization. There are typically five veterinarians on staff with new veterinarians trained to use the coding system. Difficult cases are discussed amongst the team of veterinarians with an agreed upon final diagnostic code.

Distraction was assessed through evaluations during the training period. There are currently 31 dog trainers who are similarly trained to use a coding system with defined criteria for standardized trait characterization. Trainer identification is recorded along with the scoring of individual dog performance at each time point. Dogs return from puppy raisers and start specialized training at 14–16 months of age. The mid-term blindfold test typically occurs 6–8 weeks after the start of training, while the final blindfold test is conducted 12–14 weeks into training. Distractibility was initially rated on a categorical scale but was subsequently converted to a binary classification of ‘high’ and ‘low’ for this study due to ninety-three percent of animals being categorized into only two groups. This avoids overfitting the models to skewed ordinal data and allows for a more robust comparison to be made between all binary traits. One hundred and fifteen instructors scored dogs during the 26-year period.

For health traits, the sample consisted of 1188 Labrador Retrievers, 482 German Shepherds, 239 Golden Retrievers, and 111 Labrador and Golden Retriever crosses. For distraction, there were 847 Labrador Retrievers, 257 German Shepherds, 125 Golden Retrievers, and 61 Labrador and Golden Retriever crosses in the dataset. The case counts for each trait are listed below (Table 1).

### 2.3. Genotypic Data

The Seeing Eye routinely collects whole-blood samples from all their dogs for DNA extraction and storage in their own biobank. They perform DNA extraction in house and create genotype subsets of dogs annually. Additional archived blood samples were extracted for this study and similarly genotyped. DNA extraction protocols in both laboratories followed general Qiagen (Qiagen, Germantown, MD, USA) PureGene Extraction protocols with in-house buffers. Quality control and quantification procedures ensured the integrity of the extracted DNA and compliance with genotyping standards. Genotyping was performed using three different single-nucleotide polymorphism (SNP) chips: the EMBARK (Embark Veterinary Inc., Boston, MA, USA) panel, a 220k Illumina (Illumina Inc., San Diego, CA, USA) microarray chip, and a 173k Illumina chip. The markers in the panel were mapped to CanFam 3.1 [33].

These datasets were merged, resulting in a comprehensive dataset of 239,478 SNPs across 2176 samples, with a genotyping rate of 91.4%. The data were filtered for genotype call rate and sample call rate genotypes (<90%), Hardy Weinberg equilibrium (*p* < 1 × 10^−5^), and minor allele frequency (<0.01) using PLINK ver 1.9 [34]. Post quality control, 166,463 SNPs and 2176 dogs were available for imputation. Imputation was performed following the method previously described in [35]. Briefly, the genotypes were phased and imputed using a reference panel mapped to CanFam 3.1 [33] comprising 660 dogs across 157 modern breeds and village dogs, including 15 German Shepherds, 20 Golden Retrievers, and 23 Labrador Retrievers [36]. We used Beagle ver 5.2 for both phasing and imputation [37,38]. Imputation was performed on a per-chromosome basis using mostly default settings; however, the effective population size (*ne*) was set to 200 based on previous work [39]. The imputed dataset included 10 dogs for which whole-genome sequence data were available. The accuracy of imputation was assessed based on genotype concordance and imputation quality scores [40]. Variants with a Beagle Dosage R-squared (DR2) ≥ 0.6 were retained for downstream analyses. The average genotype concordance rate between the imputed and true genotypes post DR2 filtering was 96.2%. The dataset was further pruned for LD, with PLINK ver 1.9 [34], using the -indep-pairwise flag (window size of 50 SNPs, step size of 5, and r2 threshold of 0.7), resulting in a final dataset of 1,219,623 SNPs. The dataset was then parsed to the 220 k SNPs present on the Illumina chip and then a random 50 k further subsample. The SNP density datasets, consisting of 1.2 million, 220 K, and 50 K SNPs, were selected to represent varying levels of marker densities commonly used in livestock genomic prediction. The 1.2 million SNP dataset reflects the potential of emerging, cost-effective, low-pass sequencing technologies to capture much higher density data compared to traditional SNP chips. Both smaller datasets correspond to existing work related to livestock, exploring similar densities’ effects on model performance.

### 2.4. Genomic Prediction Models

Genomic Best Linear Unbiased Prediction (GBLUP): The genomic relationship matrix was constructed using SNP data to capture genetic relationships among individuals. Analyses and matrix construction were performed using SNP & Variation Suite v8.9.1 (Golden Helix, Inc., Bozeman, MT, USA, www.goldenhelix.com, accessed on 6 December 2024). Pseudo-heritability was calculated from a full-population GBLUP analysis of each trait. The restricted maximum likelihood (REML) algorithm used was Efficient Mixed-Model Association (EMMA).

All machine learning models were evaluated using the area under the receiver operating characteristic curve (AUROC) to optimize hyperparameters, as it allows for assessment even with potential low sensitivity issues in overtrained data. Hyperparameters were optimized with a random search followed by a grid search of the characteristics of the top-performing versions of the models.

Random Forest (RF): Various hyperparameters, including n estimators, max features, max depth, min samples split, min samples leaf, bootstrap, criterion, and class weight, were optimized. This analysis was performed using the scikit-learn v1.5.1 Python package [32].

Support Vector Machine (SVM): Various kernel functions (linear, polynomial, radial basis function) were tested. Model training involved optimizing hyperparameters such as C, kernel, gamma, degree, coef0, and shrinking probability. This analysis was performed using the scikit-learn v1.5.1 Python package [32].

Extreme Gradient Boosting (XGB): The XGB model underwent iterative training with hyperparameter tuning focusing on n estimators, max depth, learning rate, subsample, col sample by tree, gamma, lambda, alpha, scale pos weight, objective, and eval metric. This analysis was performed using the XGBoost v2.1.1 Python package [41].

Multilayer Perceptron (MLP): Neural network hyperparameters, including hidden layer sizes, activation, solver, alpha, batch size, learning rate, learning rate init, max iter, tol, momentum, and early stopping, were optimized. This analysis was performed using the scikit-learn v1.5.1 Python package [32].

### 2.5. Model Validation and Evaluation

Models were evaluated using 5-fold cross-validation, which involved partitioning the data into five subsets, training on four subsets and validation on the remaining subset. This ensured a comprehensive evaluation and allowed each individual to be in the validation dataset once. Performance metrics were calculated by comparing predicted phenotypes to true values from the run when a given individual was in the validation dataset.

AUROC Calculation: The area under the receiver operating characteristic curve (AUROC) measures the ability of a model to distinguish between classes. It is calculated by plotting the true positive rate (sensitivity) against the false positive rate (1—specificity) at various threshold settings. The AUROC score ranges from 0 (completely incorrect) to 0.5 (no discrimination) to 1 (perfect discrimination).

MCC Calculation: The Matthews correlation coefficient (MCC) is used to assess the quality of binary classifications, especially in the presence of imbalanced datasets. It considers true and false positives and negatives and is calculated using Equation (1).MCC = (TP × TN − FP × FN)/√((TP + FP) × (TP + FN) × (TN + FP) × (TN + FN))(1)

TP is the number of true positives, TN is the number of true negatives, FP is the number of false positives, and FN is the number of false negatives. The values for the MCC range from −1 to 1, with 0 representing a naïve model.

## 3. Results

The traits varied in heritability among each other and when calculated from the different breed groups and SNP densities (Figure 1). Distichiasis had the highest pseudo-heritability calculated from the Genomic Best Linear Unbiased Prediction (GBLUP) analysis in the all-breed group (0.33–0.34) and Labrador/Golden Retriever (LR/GR) group (0.30–0.31). Within breeds, Golden Retrievers (GRs) had a higher heritability (0.18–0.19) than Labrador Retrievers (LRs) (0.09–0.11), and German Shepherds (GS) had no cases present. Anodontia was similarly heritable in the all-breed, GS, LR, and LR/GR breed groups (0.22–0.27) but had a heritability estimate of zero in the GR group, possibly due to low case numbers. Oral papillomatosis had a low heritability of 0.04–0.06 across all groups. Distraction was slightly more heritable among all groups (0.13–0.16) except the GR group, which had a lower heritability. There was no difference in the pseudo-heritabilities across the single-nucleotide polymorphism (SNP) datasets for any of the traits.

The trait with the highest overall performance was distichiasis (Figure 2), which had the highest heritability among the four traits assessed, ranging from 0.33–0.34 in the all-breed group to lower values within individual breed groups (GR: 0.18–0.19; LR: 0.09–0.11). The total number of cases included 116 dogs: 93 GR, 18 LR, 5 LR/GR, and 0 GS cases. There was no significant difference in model performance within each breed group/SNP dataset, but Extreme Gradient Boosting (XGB) had the highest average Matthews correlation coefficient (MCC) (0.25), while GBLUP and Random Forest (RF) had the highest area under the receiver operating characteristic curve (AUROC) across the dataset (0.81). Comparing across breeds, the all-breed and LR/GR groups had the highest performance, with an average MCC of 0.37 and 0.33 and AUROCs of 0.90 and 0.89, respectively. The GR group was next with an MCC of 0.16 but a lower AUROC of 0.63. No model was able to perform highly among the LR group, and there were no cases among the GS group. There was a slight decline in the average MCC as the SNP density decreased but no change in the average AUROC.

In the case of anodontia, which had a heritability ranging from 0.22 to 0.27 in the all-breed, GS, LR, and LR/GR breed groups but had a heritability estimate of zero in the GR group, there was no significant difference in model performance within each breed group/SNP dataset (Figure 3). The total number of cases of anodontia included 272 dogs, with 17 GR, 193 LR, 16 LR/GR, and 46 GS cases. Support Vector Machine (SVM) outperformed all other models with an average MCC of 0.13 across the anodontia groups. The lowest performing models were RF and GBLUP with average MCCs of 0.07. However, Multilayer Perceptron (MLP) and RF had the lowest average AUROCs of 0.62 and 0.65, while GBLUP had the highest average AUROC of 0.68. The breed group with the highest average MCC was LR at 0.17, followed by LR/GR at 0.15 and the all-breed group at 0.13. Neither the GS nor GR group had high-performing models by either metric, with the GR group having a heritability estimate of zero and the GS group having a heritability similar to other breeds despite lower performance. There was no change in the MCC or AUROC across the SNP data types.

For oral papillomatosis, which had a low heritability of 0.04–0.06 across all groups, there was no significant difference in model performance within each breed group/SNP dataset (Figure 4). The total number of cases included 216 dogs, with 193 LR, 43 GS, 28 GR, and 20 LR/GR cases. This trait exhibited the lowest average performance across all traits, with no standout model among any breed group by MCC or AUROC. Additionally, there was no increase or decrease in performance between the SNP density groups.

For the behavioral trait, distraction, which had a heritability ranging from 0.13 to 0.16 across most groups but lower in the GR group, there was no significant difference in model performance within each breed group/SNP dataset (Figure 5). The total number of high-distraction cases included 588, with 364 LR, 126 GS, 71 GR, and 27 LR/GR cases. MLP had the lowest average MCC of 0.03, while RF, XGB, and GBLUP averaged at 0.12. All models, apart from MLP, had similar AUROC scores as well. The models performed best by MCC for the LR (0.15) and all-breed (0.13) datasets, followed by the LR/GR group (0.11), and struggled in the GR (0.05) and GS (0.04) groups. The AUROC scores followed the same trend, with the top three having AUROCs of 0.58–0.60, while the lower ones ranged between 0.52 and 0.53. There was a slight dip in the MCC from 1.2 million SNPs to 220k SNPs, with averages decreasing from 0.11 to 0.08, but the average MCC increased to 0.09 at 50 k SNPs. The AUROC also remained consistent between 0.56 and 0.57.

## 4. Discussion

This study compared the performance of Genomic Best Linear Unbiased Prediction (GBLUP) with several machine learning (ML) models—Random Forest (RF), Support Vector Machine (SVM), Extreme Gradient Boosting (XGB), and Multilayer Perceptron (MLP)—in predicting binary health traits and a behavioral trait in guide dogs. The traits analyzed included distichiasis, anodontia, oral papillomatosis, and distractibility, demonstrating varying degrees of predictive success across the different models and datasets. The performance metrics used, Matthews correlation coefficient (MCC) and area under the receiver operating characteristic curve (AUROC), offer different perspectives on model effectiveness. While the AUROC measures a model’s ability to distinguish between classes across all threshold levels, the MCC provides a balanced evaluation that accounts for true and false positives and negatives, making it particularly informative for imbalanced datasets. The differences observed between the MCC and AUROC across traits and breed groups indicate that the MCC may be more sensitive to class imbalance and prevalence rates, affecting the interpretation of predictive success in different scenarios. This underscores the importance of considering multiple performance metrics when evaluating model performance for various traits and populations.

Machine learning approaches are continuing to be investigated with various levels of success but continual improvement in efficiency, model architectures, and increased research offer a promising future. For their usage in breeding values, ML approaches have been found to perform comparably or even outperform other methods such as GBLUP depending on specific usage and datasets [14,16,17,42]. Both an advantage and drawback for the adoption of ML algorithms in various breeding programs is the documented diversity in model performance [43]. With the variety of models and their different strengths and weaknesses, finding a single robust model that performs well across many different phenotypes and reference populations remains a challenge. Another avenue of ML adoption includes deep learning approaches, although the “black box” nature of deep learning algorithms can make interpreting the results in genomic datasets more difficult [44].

Additionally, environmental factors such as training, diet, and general environmental exposure can influence health and behavior traits. The data included in this study mitigated environmental influence on model performance by using only dogs from The Seeing Eye which had very similar breeding, nutrition, training, and health evaluations. However, there was still variation across the puppy raiser households which reflects the variables affecting pet dogs. The goal of this study was to demonstrate the feasibility and value of genomic prediction in a population with reliable phenotypes yet common environmental exposure inside and outside of households similar to pet dogs. Identifying and incorporating environmental factors influencing traits, especially on a broader and more diversified dataset of dogs, is an important avenue for future research and application of gEBVs.

Across the models, no single approach consistently outperformed others for any specific data type, as the models largely overlapped in the ranges of their run scores for both MCC and AUROC. This finding aligns with previous research that utilized GBLUP as a benchmark, confirming its robustness [45]. Considering the lack of parameter optimization, which reduces steps in breeding value creation, along with its high performance, GBLUP appears to be well suited for estimating binary trait breeding values in small, closely related populations of working dogs, as present in this study.

Additionally, there was no discernible trend of performance increase or decrease in either the MCC or AUROC with different single-nucleotide polymorphism (SNP) densities. This observation supports previous studies demonstrating that lower density SNP datasets can still effectively construct breeding values in poultry and cattle [22,23,24]. For canine breeding values, this suggests that lower cost SNP chips may suffice, negating the necessity for imputation to larger datasets for binary health traits.

Distichiasis exhibited the highest overall predictive performance and pseudo-heritability. Both GBLUP and RF models achieved the highest AUROC of 0.81, while Extreme Gradient Boosting (XGB) obtained the highest average MCC of 0.25 across the breed groups. Predictive accuracy varied among breeds, with the all-breed and Labrador Retriever/Golden Retriever groups achieving AUROCs of 0.90 and 0.89, respectively. This is consistent with prior research highlighting a significant genetic component for distichiasis, particularly in breeds such as the English Cocker Spaniel and Havanese [27,28]. The high pseudo-heritability and model performance within the breeds in the present study may be attributed to the close lineage of individuals exhibiting the trait. Interestingly, while Labrador Retrievers had a lower pseudo-heritability compared to Golden Retrievers, the combination of Labradors and Golden Retrievers outperformed both individual groups based on the MCC and AUROC. Additionally, the all-breed group had the highest pseudo-heritability and correspondingly higher scores for the MCC and AUROC. This highlights the importance of reference population construction. The increased performance among the mixed-breed groupings may be due in part to breed-specific markers being used to exclude dogs from the breeds with low to no cases, which can artificially increase the model’s performance. As such, understanding the breeds and prevalence rates can help in identifying possible inflation in reported model performance. In smaller and highly homogeneous breed groups, exploring breed-specific variants could further refine predictions, but it also raises the risk of overfitting if the variants are limited to very few breeding lines. While adding dogs, particularly increasing the number of cases, can increase the external validity, introducing dogs without the uniform phenotypic and genotypic data in the present population introduces additional variation into the theoretical dataset and may alter internal validity.

Anodontia and distractibility displayed higher predictive performance in some breed groupings compared to others, with Golden Retrievers showing the lowest pseudo-heritability, MCC, and AUROC for both traits. This may be due to the fact that the Golden Retriever group was the smallest, comprising 125 individuals in the distractibility dataset and 239 in the health traits dataset. This follows the trend present in distichiasis, where the Golden Retriever group had similarly lower predictive accuracy. Across all other groups, the pseudo-heritabilities and performance metrics were similar within each trait. This is despite the variation in group size and case counts between the Labrador Retrievers and German Shepherds. This result may suggest that Golden Retrievers may have either breed-specific variants that do not allow for as accurate predictions or that a threshold of population size is needed for the model to reach a plateau in performance. Possible explanations that may also be at play include the breed-specific variants having more complex polygenic interactions and only being identifiable with sufficient power among the specific population, in this case Golden Retrievers.

Conversely, oral papillomatosis offered an opportunity to investigate the effects of model and dataset on performance for low-heritability traits. However, as none of the models or datasets exhibited significant differences, this suggests that model selection may have less impact on predictive performance for traits with low heritability. Consequently, a standardized modeling approach may be sufficient for estimating breeding values in such cases, regardless of the specific heritability of the binary health trait.

The variability in breed composition, genetic diversity, and trait expression within the reference population illustrates the challenges inherent in constructing breeding values for functional applications. As genomic breeding values are developed for dogs, careful consideration must be given to the groups on which the models are trained. Working dogs raised in a single location offer consistency and standardized phenotypes; however, this may lead to diminished external validity due to the homogeneity of these closed populations compared to the broader breed population.

Future directions for breeding values in dogs must address unique challenges, including the lack of uniform selection criteria across different breeds and breeders. Standardization suggestions have been proposed, and working dog colonies may serve as ideal testing grounds for these approaches [46]. Additionally, the fragmented nature of dog breeding, with many breeders working with small numbers of animals with unique selective goals, may pose a challenge. The lack of a large overseeing authority akin to the Council on Dairy Cattle Breeding (CDCB) in dairy cattle to standardize breeding goals and phenotypes and collect both phenotypic and genotypic datasets presents another significant hurdle to standardization attempts. Although both the American Kennel Club (AKC) and the International Working Dog Registry have the potential to fill this gap, they currently lack large reference populations with genotypic and phenotypic data. Finally, as evidenced by the variation in the results across breed in the present study, reference populations must be carefully constructed with target populations in mind when attempting to construct a functional breeding value.

Our findings demonstrate that GBLUP, along with ML models, can effectively predict both health and behavioral traits in guide dogs, suggesting that genomic estimated breeding values (gEBVs) are valuable tools in selection programs. By enabling breeders to make informed decisions based on genetic potential, gEBVs can accelerate genetic progress and improve the overall health and performance of dog populations. Furthermore, our comparative analysis revealed that no single model consistently outperformed others across different traits and breeds, highlighting the robustness of GBLUP due to the absence of hyperparameter optimization along with the potential flexibility in model choice depending on specific breeding objectives. These results emphasize the practicality of incorporating gEBVs into canine breeding strategies, especially when considering traits of varying heritabilities and prevalence. The results also provide a proof of concept for the creation of multi-trait indices to capture the genomic influence in a series of related traits. Despite some challenges, there remains substantial potential for improvement in the current landscape. Advancements in validated health trait breeding values and the development of selection indices incorporating multiple traits could significantly enhance the utility of breeding values in dogs.

## 5. Conclusions

In conclusion, this study demonstrates that genomic prediction models—including Genomic Best Linear Unbiased Prediction (GBLUP) and machine learning approaches like Random Forest (RF), Support Vector Machine (SVM), Extreme Gradient Boosting (XGB), and Multilayer Perceptron (MLP)—are effective tools for predicting breeding values for both health and behavioral traits in guide dogs. By evaluating these models across different breeds, traits with varying heritabilities, and SNP marker densities, we found that all models performed similarly, with no single model consistently outperforming the others. Notably, GBLUP emerged as the most logistically efficient model for breeders to quickly master and implement (due to the lack of hyperparameter optimization), making it a practical choice for canine breeding programs.

These findings suggest that lower density SNP datasets are sufficient for constructing accurate genomic estimated breeding values (gEBVs), potentially reducing the costs associated with high-density genotyping. This is particularly significant for breeding programs with limited resources. By enabling breeders to make more informed selection decisions based on genetic potential, the incorporation of genomic prediction models can accelerate genetic progress and improve the overall health and performance of dog populations. Future research should focus on standardizing phenotypic assessments and expanding reference populations to enhance the utility and applicability of genomic selection in canine breeding, bridging the gap between dogs and agricultural species in genetic breeding practices.

## Figures and Tables

**Figure 1 animals-15-00408-f001:**
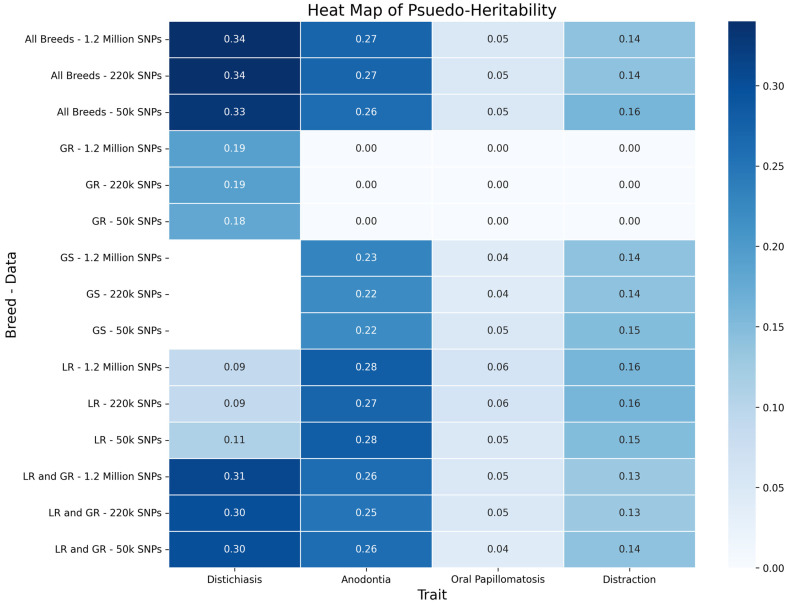
Heat Map of pseudo-heritability of traits for different breed datasets (all breeds (includes all dogs regardless of breed), GR = Golden Retriever, GS = German Shepherd, LR = Labrador Retriever, and LR/GR (includes all LRs, GRs, and LR/GR crosses)) and SNP density datasets. Heritabilities were calculated from GBLUP analysis of each trait organized by breed and dataset.

**Figure 2 animals-15-00408-f002:**
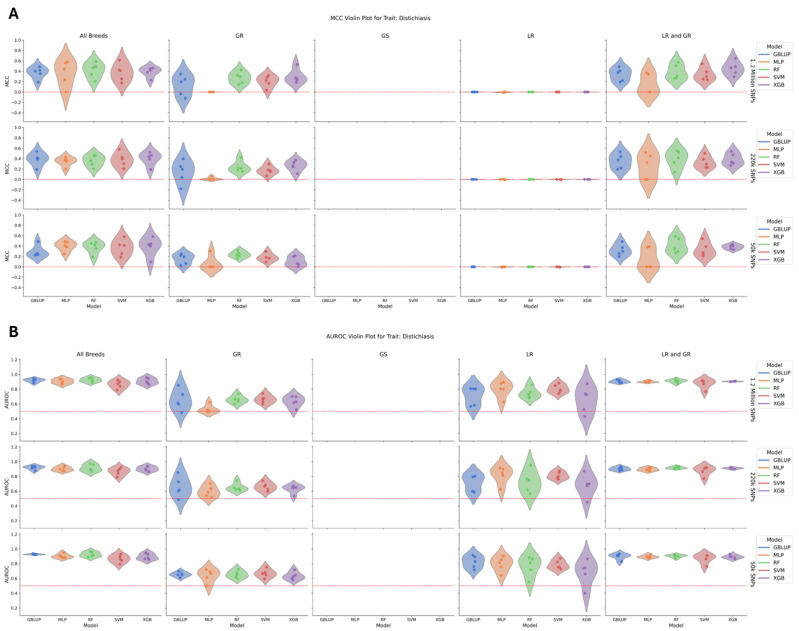
(**A**) A violin plot of model performance by MCC for distichiasis. The data are partitioned by SNP dataset in the rows and breed group in the columns with the models colored as follows: GBLUP—blue, MLP—orange, RF—green, SVM—red, and XGB—purple. The red dotted line represents the expected performance of a totally naïve model with an MCC of 0. There was no significant difference in model performance within each breed group/SNP dataset. (**B**) A violin plot of model performance by AUROC for distichiasis. The data are partitioned by SNP dataset in the rows and breed group in the columns with the models colored as follows: GBLUP—blue, MLP—orange, RF—green, SVM—red, and XGB—purple. The red dotted line represents the expected performance of a totally naïve model with an AUROC of 0.5. There was no significant difference in model performance within each breed group/SNP dataset.

**Figure 3 animals-15-00408-f003:**
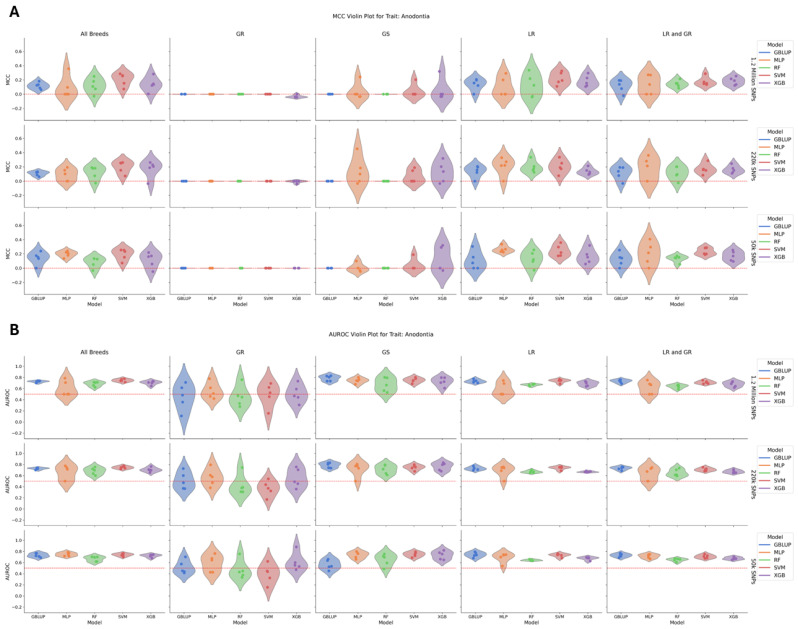
(**A**) A violin plot of model performance by MCC for anodontia. The data are partitioned by SNP dataset in the rows and breed group in the columns with the models colored as follows: GBLUP—blue, MLP—orange, RF—green, SVM—red, and XGB—purple. The red dotted line represents the expected performance of a totally naïve model with an MCC of 0. There was no significant difference in model performance within each breed group/SNP dataset. (**B**) A violin plot of model performance by AUROC for anodontia. The data are partitioned by SNP dataset in the rows and breed group in the columns with the models colored as follows: GBLUP—blue, MLP—orange, RF—green, SVM—red, and XGB—purple. The red dotted line represents the expected performance of a totally naïve model with an AUROC of 0.5. There was no significant difference in model performance within each breed group/SNP dataset.

**Figure 4 animals-15-00408-f004:**
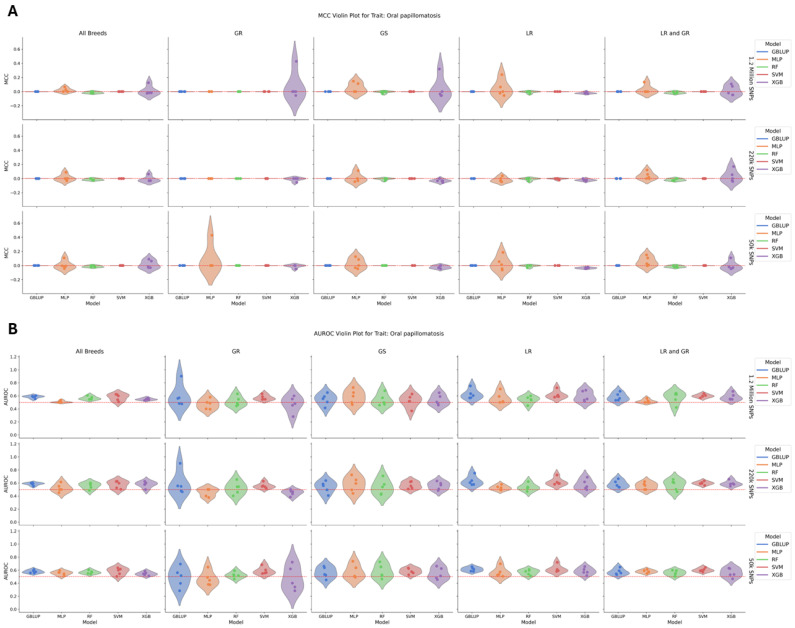
(**A**) A violin plot of model performance by MCC for oral papillomatosis. The data are partitioned by SNP dataset in the rows and breed group in the columns with the models colored as follows: GBLUP—blue, MLP—orange, RF—green, SVM—red, and XGB—purple. The red dotted line represents the expected performance of a totally naïve model with an MCC of 0. There was no significant difference in model performance within each breed group/SNP dataset. (**B**) A violin plot of model performance by AUROC for oral papillomatosis. The data are partitioned by SNP dataset in the rows and breed group in the columns with the models colored as follows: GBLUP—blue, MLP—orange, RF—green, SVM—red, and XGB—purple. The red dotted line represents the expected performance of a totally naïve model with an AUROC of 0.5. There was no significant difference in model performance within each breed group/SNP dataset.

**Figure 5 animals-15-00408-f005:**
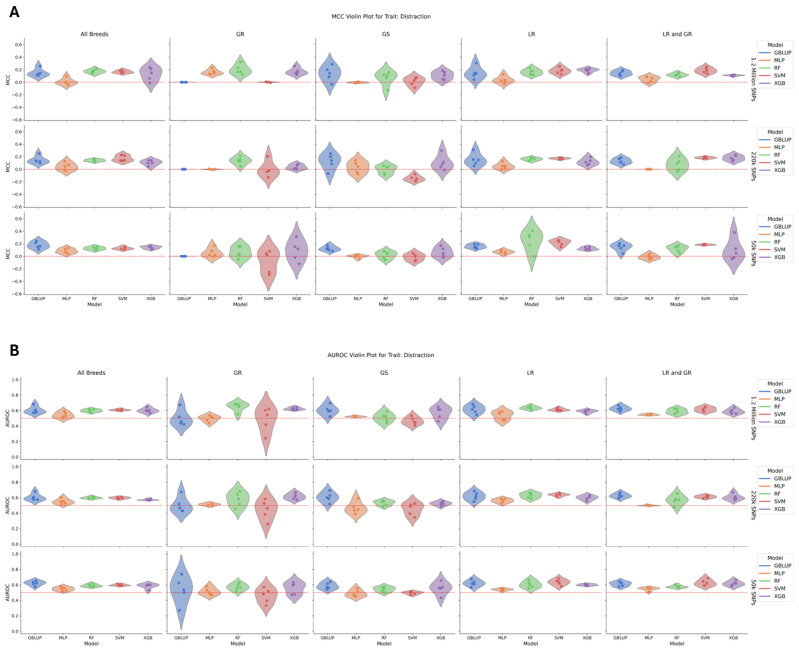
(**A**) A violin plot of model performance by MCC for distraction. The data are partitioned by SNP dataset in the rows and breed group in the columns with the models colored as follows: GBLUP—blue, MLP—orange, RF—green, SVM—red, and XGB—purple. The red dotted line represents the expected performance of a totally naïve model with an MCC of 0. There was no significant difference in model performance within each breed group/SNP dataset. (**B**) A violin plot of model performance by AUROC for distraction. The data are partitioned by SNP dataset in the rows and breed group in the columns with the models colored as follows: GBLUP—blue, MLP—orange, RF—green, SVM—red, and XGB—purple. The red dotted line represents the expected performance of a totally naïve model with an AUROC of 0.5. There was no significant difference in model performance within each breed group/SNP dataset.

**Table 1 animals-15-00408-t001:** Counts of cases for the behavior and health traits by breed.

Trait	Total Cases	Labrador Retrievers	German Shepherds	Golden Retrievers	Labrador/Golden Retriever Crosses
Distichiasis	116	18	0	93	5
Anodontia	272	193	46	17	16
Oral Papillomatosis	216	125	43	28	20
High Distractibility	588	364	126	71	27

## Data Availability

The datasets presented in this article are not readily available as they are the private ownership of The Seeing Eye. Author, Katy M. Evans, may be contacted to inquire about the datasets. The script for the machine learning models is available on google colab: https://colab.research.google.com/drive/1CTYO6ly20ZpTwLsJGXace4KYbBXZL7gP?usp=sharing (accessed on 6 December 2024).

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
