# Peer review of "Performance Comparison of Genomic Best Linear Unbiased Prediction and Four Machine Learning Models for Estimating Genomic Breeding Values in Working Dogs"

_animals, 2025, doi:10.3390/ani15030408_

Round 1

Reviewer 1 Report

Comments and Suggestions for Authors

The paper presents a solid foundation on genomic prediction techniques and their application in breeding programs, particularly highlighting the use of models such as GBLUP, Random Forest, SVM, XGB, and MLP. The detailed explanation of the advantages of each method and the historical success in agricultural species is relevant to contextualize the use of these tools in dogs.

However, the justification for the study lacks a clearer focus on the practical importance and direct impact of the work. The introduction is overly focused on describing the models and their technical characteristics, relegating the relevance of the studied traits (anodontia, distichiasis, oral papillomatosis, and distraction) for the genetic improvement of dogs to the background.

The methodology, although consistent with the study's objectives, presents some gaps that could compromise the robustness, replicability, and, consequently, the scientific impact and relevance of the work. The study mentions crosses between Labradors and Golden Retrievers, followed by backcrossing with Labradors, but does not justify this strategy, which impairs the interpretation of genetic and phenotypic results. Moreover, the lack of information about the age of the dogs and the duration of the training limits the understanding of the role of development in the formation of the behaviors analyzed, affecting the study's applicability and replicability.

Another point to consider is the transformation of the behavioral scale into a binary classification (‘high’ and ‘low’) without sufficient justification, which may result in an excessive simplification of a complex behavior, compromising the accuracy of the results and the extrapolation to other populations. The absence of mention regarding the number of evaluators and the standardization of distraction classification criteria increases the risk of subjectivity, undermining the reliability of the findings and the scientific validity of the study.

Although the study identified interesting variations in the heritability of traits and highlighted distichiasis as the trait with the best performance, there are opportunities to improve the accuracy of the models, especially in breeds with limited data, such as Golden Retrievers and German Shepherds. Increasing the number of cases, exploring more complex modeling techniques, such as neural networks, and performing a deeper analysis of SNP density could improve model performance, particularly for traits with low heritability, such as oral papillomatosis and distraction.

The discussion is well-founded and provides a good overview of the effectiveness of genomic prediction models and machine learning in the context of guide dog breeding. The detailed analysis of the results and practical implications is valuable, but it would be interesting to explore more deeply the study's limitations, such as the possible lack of representativeness of the included breeds and the limitations of machine learning models when applied to genetic data. Additionally, a more detailed analysis of how external variables such as environment, diet, and training can influence behavioral and health traits would be relevant to provide a more comprehensive perspective on the applicability of the models.

The discussion could also explore more deeply how population size and genetic diversity influence predictions, especially for traits with low prevalence, such as oral papillomatosis. Although breed performance was discussed, a deeper analysis of how specific genetic variants of each breed influence the results would be useful, especially in small and homogeneous working dog populations.

This study demonstrates that genomic prediction models—including Genomic Best Linear Unbiased Prediction (GBLUP) and machine learning approaches such as Random Forest (RF), Support Vector Machine (SVM), Extreme Gradient Boosting (XGB), and Multilayer Perceptrons (MLP)—are effective tools for predicting breeding values of health and behavior traits in guide dogs. When evaluating these models in different breeds, traits with varying heritabilities, and SNP marker densities, the results showed that all models had similar performances, with no single model consistently standing out.

GBLUP emerged as the most computationally efficient model, due to the lack of hyperparameter optimization, making it a practical choice for canine breeding programs. These findings suggest that low-density SNP datasets are sufficient to construct accurate genomic estimated breeding values (gEBVs), which could reduce costs associated with high-density genotyping—a particularly relevant point for programs with limited resources.

Although genomic prediction models like GBLUP and machine learning approaches have been widely evaluated in other species and shown effectiveness in prediction, for the study to have a greater scientific impact and innovation, improvements to the models would be necessary.

Comments on the Quality of English Language

The English in the article is good, but there are some points that can be adjusted to improve clarity and fluency. Here are some revision suggestions:

  • Some sentences are quite long and can be divided to improve comprehension.
  • Make sure to use technical terms consistently throughout the text, such as using "machine learning" instead of "aprendizado de máquina" if the goal is to maintain English terminology.
  • There are some sentences that can be restructured for greater clarity and fluency.

Author Response

For research article: Performance comparison of GBLUP and four machine learning models for estimating genomic breeding values in working dogs

Response to Reviewer 1 Comments

1)   Summary:

Author Response: Thank you for reviewing our manuscript and providing valuable feedback to improve our submission.  Please find the detailed responses below and the corresponding revisions/corrections highlighted in track changes in the re-submitted files. Please note that the line references are correct if you are reviewing the manuscript in “Simple Markup” of tracked changes.

2)   General Evaluation:

( ) The quality of English does not limit my understanding of the research.
(x) The English could be improved to more clearly express the research.

  •  

Yes

Can be improved

Must be improved

Not applicable

Does the introduction provide sufficient background and include all relevant references?

( )

( )

(x)

( )

Is the research design appropriate?

( )

( )

(x)

( )

Are the methods adequately described?

( )

( )

(x)

( )

Are the results clearly presented?

( )

(x)

( )

( )

Are the conclusions supported by the results?

( )

( )

(x)

( )

3)   Comments and Suggestions for Authors:

Comment 1: “The paper presents a solid foundation on genomic prediction techniques and their application in breeding programs, particularly highlighting the use of models such as GBLUP, Random Forest, SVM, XGB, and MLP. The detailed explanation of the advantages of each method and the historical success in agricultural species is relevant to contextualize the use of these tools in dogs.

However, the justification for the study lacks a clearer focus on the practical importance and direct impact of the work. The introduction is overly focused on describing the models and their technical characteristics, relegating the relevance of the studied traits (anodontia, distichiasis, oral papillomatosis, and distraction) for the genetic improvement of dogs to the background.”

Response 1: We appreciate your point and have added edits on lines 32-35, 72-84, 147-151 to address this concern by more explicitly highlighting the practical significance and real-world impact of the research along with discussions on the effective and ethical usage of breeding values. The revised text clarifies how these genomic prediction methods can mitigate health and behavioral problems in dogs—leading to reduced discomfort, enhanced working performance, and fewer animals being rehomed.  With that said, the focus of this paper is on comparing the models. The traits we used, while impacting animal health and guide dog success, were chosen to test aspects of the model performance which is why we highlight the models more so than the value of these particular traits.  They are indeed important to guide dog success and dog health and we’ve highlighted this in response to your comment.  We are planning a separate publication that will focus on generating gEBVs for a much broader array of traits using a single prediction model.  This next publication will highlight the importance of the traits much more so than this publication which focuses more on the models.

Comment 2: “The methodology, although consistent with the study's objectives, presents some gaps that could compromise the robustness, replicability, and, consequently, the scientific impact and relevance of the work. The study mentions crosses between Labradors and Golden Retrievers, followed by backcrossing with Labradors, but does not justify this strategy, which impairs the interpretation of genetic and phenotypic results. Moreover, the lack of information about the age of the dogs and the duration of the training limits the understanding of the role of development in the formation of the behaviors analyzed, affecting the study's applicability and replicability.”

Response 2: Additions were added to lines 191-195 explaining the rationale behind introducing and backcrossing Labrador–Golden Retriever crosses, clarifying how these strategies combine key working traits and increase genetic diversity while preserving a consistent phenotype. The duration of the training and information on phenotypic collection is in lines 198-199, 209-212, and 220-223.  

Comment 3: “Another point to consider is the transformation of the behavioral scale into a binary classification (‘high’ and ‘low’) without sufficient justification, which may result in an excessive simplification of a complex behavior, compromising the accuracy of the results and the extrapolation to other populations. The absence of mention regarding the number of evaluators and the standardization of distraction classification criteria increases the risk of subjectivity, undermining the reliability of the findings and the scientific validity of the study.”

Response 3: Text currently exists describing that 93% of the dogs evaluated for distractibility were classified into only two of the categories for distraction.  To elaborate on the transition from ordinal to binary, text was added to lines 223-227 explaining the advantages of avoiding a skewed dataset in a regression model, and increased interpretability of utilizing a binary dataset for this trait which complements the health traits which are also binary.  We have also added details on the number of veterinarians and trainers in The Seeing Eye program and their training to use an established coding system for each trait (lines 212-216, 217-220, and 227-228).

Comment 4: “Although the study identified interesting variations in the heritability of traits and highlighted distichiasis as the trait with the best performance, there are opportunities to improve the accuracy of the models, especially in breeds with limited data, such as Golden Retrievers and German Shepherds. Increasing the number of cases, exploring more complex modeling techniques, such as neural networks, and performing a deeper analysis of SNP density could improve model performance, particularly for traits with low heritability, such as oral papillomatosis and distraction.”

Response 4: Agreed. The case prevalence in the population is from animals present in the population within The Seeing Eye to explore the effectiveness of breeding values on a closed breeding population of working dogs. While additional animals can increase the external validity, introducing animals without the uniform phenotypic and genotypic data in the present population introduces additional variation into the theoretical dataset and may alter internal validity. We have added text to the discussion that addresses this and comment 6 below (lines 506-513, 524-527). More animals, including cases, will be added by The Seeing Eye and used for gEBV generation in the future but were unavailable for this publication.

For the increase in model complexity, existing studies discussed in lines 110-142 have shown that the models included have sufficient complexity from exploration in other species. The ease of use and existing body of literature allows for a more practical suggestion for further researchers and breeding managers.

In regard to the suggestion of including a neural network to the analysis, the included MLP model is a neural network and, in the manuscript, lines 127-129 state “Multilayer Perceptrons (MLP) are neural network models inspired by the structure and function of the human brain. MLPs consist of multiple layers of interconnected nodes that can model complex, nonlinear relationships within the data”

To better address the marker density justification, text was added to lines 267-274 explaining the rational behind the choices for SNP marker densities tested.

Comment 5: “The discussion is well-founded and provides a good overview of the effectiveness of genomic prediction models and machine learning in the context of guide dog breeding. The detailed analysis of the results and practical implications is valuable, but it would be interesting to explore more deeply the study's limitations, such as the possible lack of representativeness of the included breeds and the limitations of machine learning models when applied to genetic data. Additionally, a more detailed analysis of how external variables such as environment, diet, and training can influence behavioral and health traits would be relevant to provide a more comprehensive perspective on the applicability of the models.”

Response 5: Agreed. To better elaborate on these limitations, additional text was added in lines 454-476.   We included how ML models are limited when applied to genomic data and discussed the influence of environmental factors and how they were mitigated given our use of only The Seeing Eye dogs. We also note the importance of examining the influence of environmental factors with larger, more diverse dog population datasets.

Comment 6: “The discussion could also explore more deeply how population size and genetic diversity influence predictions, especially for traits with low prevalence, such as oral papillomatosis. Although breed performance was discussed, a deeper analysis of how specific genetic variants of each breed influence the results would be useful, especially in small and homogeneous working dog populations.”

Response 6: Agreed. Increased discussion of the advantages and disadvantages of the breed specific analyses was added to lines 506-513, 524-527 to address the possible influences at play.

Comment 7: “This study demonstrates that genomic prediction models—including Genomic Best Linear Unbiased Prediction (GBLUP) and machine learning approaches such as Random Forest (RF), Support Vector Machine (SVM), Extreme Gradient Boosting (XGB), and Multilayer Perceptrons (MLP)—are effective tools for predicting breeding values of health and behavior traits in guide dogs. When evaluating these models in different breeds, traits with varying heritabilities, and SNP marker densities, the results showed that all models had similar performances, with no single model consistently standing out.

GBLUP emerged as the most computationally efficient model, due to the lack of hyperparameter optimization, making it a practical choice for canine breeding programs. These findings suggest that low-density SNP datasets are sufficient to construct accurate genomic estimated breeding values (gEBVs), which could reduce costs associated with high-density genotyping—a particularly relevant point for programs with limited resources.

Although genomic prediction models like GBLUP and machine learning approaches have been widely evaluated in other species and shown effectiveness in prediction, for the study to have a greater scientific impact and innovation, improvements to the models would be necessary.”

Response 7: As previous work has focused on agricultural species, and genomic breeding values have not been constructed in a population of dogs previously, the study’s goal was to describe how existing models perform on this new dataset of animals with novel traits.  We acknowledge that increased case numbers, increased animals, differing populations, and further examination of genetic marker densities would impact and potentially improve our models.  There is much left to explore, yet two of the major challenges of establishing gEBVs for dogs is in the development of a phenotypic and genomic reference dataset and understanding of how existing models perform in such a dataset.  We have taken this first step of establishing a reference dataset and comparing model performance. There is much left to do.

Comment 8: “The English in the article is good, but there are some points that can be adjusted to improve clarity and fluency. Here are some revision suggestions:

Some sentences are quite long and can be divided to improve comprehension.”

Response 8: No specific instances were included but corrections were made on the following lines: 28-32, 60, 116, 126, 188-200.

Comment 9: “Make sure to use technical terms consistently throughout the text, such as using "machine learning" instead of "aprendizado de máquina" if the goal is to maintain English terminology.”

Response 9: Could you please identify specific instances for further clarification? We did not find any Portuguese in the manuscript.

Comment 10: “There are some sentences that can be restructured for greater clarity and fluency.”

Response 10: Similar to Comment 8 with lines 28-32, 60, 116, 126, 188-200 altered.

Thank you again for taking the time to review our manuscript.

Reviewer 2 Report

Comments and Suggestions for Authors

The relatively low estimates of heritability suggest substantial environmental variance affecting these traits. However, the description of the analyses is devoid of an mention of controlling such sources of variation (i.e., contemporary groups, trainers, etc).

Line 497: the conclusion regarding computational efficiency, while undoubtedly true, is not supported by data regarding the computational resources used in each class of evaluation. 

It would be of interest to have constructed an even less than 50K SNP chip. 

Author Response

For research article: Performance comparison of GBLUP and four machine learning models for estimating genomic breeding values in working dogs

Response to Reviewer 2 Comments

1)   Summary:

Thank you for reviewing our manuscript and providing valuable feedback to improve our submission.  Please find the detailed responses below and the corresponding revisions/corrections highlighted in track changes in the re-submitted filesPlease note that the line references are correct if you are reviewing the manuscript in “Simple Markup” of tracked changes.

2)   General Evaluation:

Quality of English Language

(x) The quality of English does not limit my understanding of the research.
( ) The English could be improved to more clearly express the research.

  •  

Yes

Can be improved

Must be improved

Not applicable

Does the introduction provide sufficient background and include all relevant references?

(x)

( )

( )

( )

Is the research design appropriate?

( )

(x)

( )

( )

Are the methods adequately described?

( )

(x)

( )

( )

Are the results clearly presented?

(x)

( )

( )

( )

Are the conclusions supported by the results?

( )

(x)

( )

( )

3)   Comments and Suggestions for Authors:

Comment 1: “The relatively low estimates of heritability suggest substantial environmental variance affecting these traits. However, the description of the analyses is devoid of an mention of controlling such sources of variation (i.e., contemporary groups, trainers, etc).”

Response 1: This is an important limitation and clarifying text on the general number of trainers and veterinarians as well as the use of an established coding system has been added to the methods (lines 212-216, 217-220, and 227-228). We also added how the influence of environmental factors is mitigated with the use of The Seeing Eye dataset but note that this is an important point to research as the dataset grows and becomes more diverse.  (lines 466-476)

Comment 2: “Line 497: the conclusion regarding computational efficiency, while undoubtedly true, is not supported by data regarding the computational resources used in each class of evaluation. “

Response 2: Agreed. Due to a lack of data on the performance metrics included across the different software packages; lines 41, 47, 480-483 were adjusted to highlight that the efficiency lies in the lack of a need for continued hyperparameter tuning and not simply increased program efficiency.  We realized after the fact that it would have been great to have actual run time data for the different programs but that was not something we recorded at the time.

Comment 3: “It would be of interest to have constructed an even less than 50K SNP chip. “

Response 3: We agree there was a lack of justification for the SNP marker densities in the original manuscript and the author’s reasoning was added in lines 267-274.  We could certainly evaluate a lower density panel but chose to have the 50K as the lowest density in our study based on technology available and much review of this parameter in livestock systems.

Thank you again for your time reviewing our manuscript.

Reviewer 3 Report

Comments and Suggestions for Authors

The paper is overall excellently written and interesting, easy to follow, and undoubtedly appropriate for the scope of the journal.
A few observations below.

Methods

202-226: This part should be better elaborated.

Firstly, you should clarify the canine reference used. If this is Illumina 220k/173k SNPs, I presume this is Canfam3.1. Were the SNPs re-mapped against a different reference (Canfam3.1 is quite old). If so, please specify. If not, I think it should be stated clearly in any case.

In the same way, this type of information should be made clear for the imputation panel.

216-217: breed of the dogs? Overall, this part should be a bit more detailed.

216: You say Beagle v5 was used – has v5 AR2 as part of the phasing output, or just DR2? 

222-225: this part should be elaborated (say, number of SNPs after fixed loci removal, why 50k etc)

Discussion

437-440: does it mean there is a threshold? Could the authors elaborate on that?

General question – the authors introduce the concept of overfitting in the introduction, should the results be discussed more under that light?

Author Response

For research article: Performance comparison of GBLUP and four machine learning models for estimating genomic breeding values in working dogs

Response to Reviewer 3 Comments

1)   Summary:

Thank you for reviewing our manuscript and providing valuable feedback to improve our submission.  Please find the detailed responses below and the corresponding revisions/corrections highlighted in track changes in the re-submitted filesPlease note that the line references are correct if you are reviewing the manuscript in “Simple Markup” of tracked changes.

2)   General Evaluation:

Quality of English Language

(x) The quality of English does not limit my understanding of the research.
( ) The English could be improved to more clearly express the research.

  •  

Yes

Can be improved

Must be improved

Not applicable

Does the introduction provide sufficient background and include all relevant references?

(x)

( )

( )

( )

Is the research design appropriate?

(x)

( )

( )

( )

Are the methods adequately described?

(x)

( )

( )

( )

Are the results clearly presented?

( )

(x)

( )

( )

Are the conclusions supported by the results?

( )

( )

( )

( )

3)Comments and Suggestions for Authors:

Comment 1: “202-226: This part should be better elaborated.

Firstly, you should clarify the canine reference used. If this is Illumina 220k/173k SNPs, I presume this is Canfam3.1. Were the SNPs re-mapped against a different reference (Canfam3.1 is quite old). If so, please specify. If not, I think it should be stated clearly in any case.

In the same way, this type of information should be made clear for the imputation panel.”

Response 1: Agreed, the reference was Canfam 3.1. We have clarified the reference genome in the manuscript on lines 247 and 255. We have also added more information on the data QC and imputation on lines 250-253 and 258-259. 

Comment 2: ”216-217: breed of the dogs? Overall, this part should be a bit more detailed.

Response 2: We have given the overall number of breeds in the imputation panel on lines 255-256. A detailed list of dogs in the imputation panel was given in our previous work, we have cited that paper.

Srikanth, K., von Pfeil, D. J., Stanley, B. J., Griffitts, C., & Huson, H. J. (2022). Genome wide association study with imputed whole genome sequence data identifies a 431 kb risk haplotype on CFA18 for congenital laryngeal paralysis in Alaskan sled dogs. Genes, 13(10), 1808.

Comment 3: 216: You say Beagle v5 was used – has v5 AR2 as part of the phasing output, or just DR2?”

Response 3: Thank you, we have corrected the mistake, Beagle v5 only outputs DR2 and the filtering was performed using DR2. (Line 264)

Comment 4: “222-225: this part should be elaborated (say, number of SNPs after fixed loci removal, why 50k etc)” 

Response 4: We agree there was a lack of justification for the SNP marker densities in the original manuscript and the author’s reasoning was added in lines 267-273. Figures were also updated to match the SNP counts present for analyses.

Comment 5: “437-440: does it mean there is a threshold? Could the authors elaborate on that?”

Response 5: A clarification to the idea is added to lines 524-527 that elaborates on the relationship between the population size and potential breed specific causative/associated variants.

Comment 6: “General question – the authors introduce the concept of overfitting in the introduction, should the results be discussed more under that light?”

Response 6: Good point. We’ve added text to the discussion on overfitting, addressing the relationship between the data and potential overfitting in lines 223-227 and 507-512.

Thank you again for your time reviewing our manuscript.